# Cryptanalysis of an Image Encryption Algorithm Based on Two-Dimensional Hyperchaotic Map

**DOI:** 10.3390/e25030395

**Published:** 2023-02-21

**Authors:** Qinmao Jiang, Simin Yu, Qianxue Wang

**Affiliations:** College of Automation, Guangdong University of Technology, Guangzhou 510006, China

**Keywords:** image encryption, hyperchaotic map, chosen plaintext attack, cryptanalysis

## Abstract

This paper analyzes the security of an image encryption algorithm based on a two-dimensional hyperchaotic map. This encryption algorithm generated chaotic sequences through a combination of two one-dimensional chaotic maps and used them as the permutation and diffusion key. Then, the image was encrypted by using the structure of row–column permutation, forward-diffusion, and backward-diffusion. The proposer claimed that the above algorithm was secure. However, it was found through cryptanalysis that the algorithm cannot withstand the chosen plaintext attack. Although the forward-diffusion and backward-diffusion of the original algorithm use two different diffusion keys and there is a ciphertext feedback mechanism, the analysis of the diffusion by iterative optimization showed that it can be equivalent to global diffusion. In addition, the generation of chaotic sequences in the encryption process is independent of the plaintext image, so the equivalent diffusion and permutation key stream can be obtained by adjusting the individual pixel values of the chosen plaintexts. Aiming at the security loopholes in the encryption algorithm, the theoretical and experimental results are presented to support the efficiency of the proposed attack and suggestions for improvement are given. Finally, compared with the performance analysis of the existing cracking algorithm, our cryptanalysis greatly improved the cracking efficiency without increasing the complexity of the data.

## 1. Introduction

With the rapid development of computer and communication technology, multimedia information, such as images, has become the main carrier of network information with its characteristics of carrying large amounts of information. The following problem is how to ensure the security of image data in the process of transmission. In recent years, due to the ergodicity, unpredictability, and initial-value sensitivity of chaotic systems [1,2,3,4], people have combined them with cryptography and gradually applied them to the field of multimedia information security, which has improved the security of multimedia information transmission to a certain extent.

Since the permutation–diffusion encryption structure was proposed, researchers have designed many novel chaotic image encryption schemes based on this structure [5,6]. For example, DNA encoding and decoding technology combined with a chaotic system was used for image encryption [7,8,9,10]; chaotic image encryption schemes based on double S-boxes were proposed [11,12,13,14]; the image compression technology was applied to chaotic image encryption [15,16,17,18]. Furthermore, in addition to designing relatively new chaotic encryption algorithms, researchers are paying more and more attention to the security of the algorithm itself. Therefore, in order to resist various attacks, various methods have been proposed to improve the security of encryption algorithms [19,20,21,22]. In [23], a dynamic key chaotic image encryption algorithm related to the plaintext was proposed, which improved the plaintext sensitivity of the encryption algorithm. In [24], a chaotic system based on a PWLCM map and an image encryption algorithm of dynamic DNA were proposed, then the chaotic sequences were used to dynamically select corresponding DNA encoding and decoding rules, making the encryption process more flexible. In [25], an image encryption scheme based on the generalized Arnold transform and RSA algorithm was proposed, which improved security by introducing the RSA algorithm. Moreover, improving new chaotic systems is also a common means [26,27]. Neural networks have been favored by academia and industry because of the diversity of their dynamics. A chaotic image encryption algorithm improved by a neural network had strong practicability in the medical field [28,29]. No matter how complex the encryption algorithm is, its purpose is to combine the chaotic system to conduct a series of permutations and diffusions of image pixels to ensure the security of image transmission.

Currently, image encryption algorithms are also becoming more and more diversified. Cryptographic designers often claim that the proposed encryption algorithm has good security based on a series of statistical test results. However, strict cryptanalysis shows that some chaotic encryption algorithms still have certain security loopholes, causing them to be cracked by attackers [30,31,32,33]. In the process of analyzing the security of the encryption algorithm, attackers try various methods to crack the algorithm and gradually develop some classic cryptanalysis methods. In [34], an encryption scheme based on a random walk matrix and a hyperchaotic system was cracked by the chosen plaintext attack method, and relevant suggestions were given for its security loopholes to improve the anti-attack capability; in [35], an image encryption algorithm combining DNA coding and a spatiotemporal chaotic system was analyzed; its equivalent key can be obtained by the chosen ciphertext attack method according to its defects, and it recovered the plaintext image; in [36], when a chaotic image encryption algorithm based on information entropy was cryptanalyzed for the effectiveness of quantifiable security measures, the differential attack method can be used to recover the equivalent key of each basic operation of the encryption algorithm.

Among many cryptanalysis methods, the chosen plaintext attack is a commonly used method. For example, the equivalent key can be obtained by selecting the corresponding special plaintext in different encryption stages for the encryption scheme of a chaotic system combined with an image block, that is the whole encryption algorithm can crack the different stages one by one [37]. Aiming at the color image encryption scheme of chaos and DNA encoding, the chosen plaintext attack could also be used to disclose its equivalent secret key [38]. In 2022, an image encryption algorithm based on a two-dimensional hyperchaotic map was proposed [39]. The encryption algorithm included two rounds of permutation and two rounds of diffusion. The permutation key and diffusion key were generated through the new coupled two-dimensional hyperchaotic map. There was a ciphertext feedback mechanism in the two-round diffusion process. According to the statistical test results, the original paper claimed that the algorithm had high security. However, the loopholes in the encryption algorithm were analyzed, and the chosen plaintext attack was used to crack it; however, the efficiency of the cracking algorithm was very low, and it took nearly 2 h to crack a 256×256 image, which obviously does not meet the actual needs [40]. By analyzing the security loopholes of the original encryption algorithm, this paper proposes a new cracking method, which greatly improves the cracking efficiency without increasing the complexity of the data.

The rest of this paper is arranged as follows: Section 2 briefly describes the image encryption algorithm proposed in [39]. In Section 3, the original encryption algorithm is analyzed as a whole, and then, the plaintext attack method is used to crack its security. In Section 4, numerical simulation experiments are carried out based on the security analysis method proposed in this paper, and the attack complexity is discussed. Finally, improvement suggestions are given for the existing security loopholes. Section 5 compares the performance of the cracking algorithm in this paper with that in [40]. The last section summarizes the content of this paper.

## 2. Description of Original Image Encryption Algorithm

This section introduces the chaotic system used in the original encryption algorithm and its specific encryption process. In [39], an image encryption algorithm based on a two-dimensional hyperchaotic map was proposed. In the original encryption algorithm, a series of chaotic sequences was generated by given initial key parameters for subsequent encryption operations based on the proposed new two-dimensional hyperchaotic system. The modified chaotic sequences were firstly used to exchange the whole row and column of the pixel value of the image and, secondly, used to carry out forward-diffusion and backward-diffusion on the pixel values to achieve the combination of chaos and the encryption algorithm.

### 2.1. Two-Dimensional Hyperchaotic System

In the original algorithm, a new 2D hyperchaotic map was proposed by using two one-dimensional chaotic maps $\sin(\frac{h\pi }{x})$ and rsin(πx), and its expression is as follows:(1)$$\begin{array}{c} \left\{\begin{array}{c} x\left(n\right)=\sin\left(\frac{h\pi }{\sin(y(n-1))}\right),\\ y(n)=r\sin(\pi x(n-1)y(n-1)), \end{array}\right. \end{array}$$
where the state variables x(n)∈(−1,1), y(n)∈(−r,r); when the control parameter is h∈[3,7], r∈[2,6], the system is in a chaotic state.

### 2.2. Description of the Original Encryption Algorithm

According to the original encryption algorithm in [39], the block diagram of image encryption is shown in Figure 1, and the detailed process is described in the following:

(1)Selection of initial key parameters:

According to Figure 1, the original encryption algorithm includes four initial key parameters, *h*, *r*, x(0), and y(0), and the values were taken as h=5, r=5, x(0)=0.5, and y(0)=0.5 for the image encryption.

(2)Generation of permutation index sequences:

Substitute the initial key parameters into Equation (Equation 1) for *m* iterations to avoid transient effects. Then, iterate it H×W more times to obtain two chaotic sequences $X={\left\{x\left(i\right)\right\}}_{i=1}^{H\times W}$ and $Y={\left\{y\left(i\right)\right\}}_{i=1}^{H\times W}$, where *H* and *W* represent the height and width of the plaintext image and N=max(H,W).

Then, the first *N* sequences of the chaotic sequences *X* and *Y* are intercepted and quantified according to Equation (Equation 2) to obtain the permutation index sequences $R={\left\{r\left(i\right)\right\}}_{i=1}^{N}$ and $T={\left\{t\left(i\right)\right\}}_{i=1}^{N}$ for image row–column permutation.
(2)$$\begin{array}{c} \left\{\begin{array}{c} r\left(i\right)=\mathrm{mod}(\mathrm{floor}(|x\left(i\right)|\times 10^{16}),\frac{H}{2}),\\ t\left(i\right)=\mathrm{mod}(\mathrm{floor}(|y\left(i\right)|\times 10^{16}),\frac{W}{2}), \end{array}\right. \end{array}$$
where floor(·) represents the rounding-down operation, $\left|\cdot \right|$ represents the absolute operation, and mod represents the modular operation.

(3)Generation of diffusion matrix:

Similarly, the chaotic sequences *X* and *Y* are quantified according to Equation (Equation 3) to obtain the integer sequences $X_{g}={\left\{x_{g}\left(i\right)\right\}}_{i=1}^{H\times W}$ and $Y_{g}={\left\{y_{g}\left(i\right)\right\}}_{i=1}^{H\times W}$, which make the values in the range of [0,255].
(3)$$\begin{array}{c} \left\{\begin{array}{c} x_{g}\left(i\right)=\mathrm{mod}(\mathrm{round}(1000(|x\left(i\right)\times 10^{16}|-\mathrm{floor}(|x\left(i\right)|\times 10^{16}))),256),\\ y_{g}\left(i\right)=\mathrm{mod}(\mathrm{round}(1000(|y\left(i\right)\times 10^{16}|-\mathrm{floor}(|y\left(i\right)|\times 10^{16}))),256), \end{array}\right. \end{array}$$
where round(·) represents the rounding operation, floor(·) represents the rounding-down operation, $\left|\cdot \right|$ represents the absolute operation, and mod represents the modular operation.

In order to facilitate subsequent diffusion operations, the sequences $X_{g}$ and $Y_{g}$ need to be written as matrices with the same size as the plaintext image matrix by scanning it in the raster order (from left to right and then from top to bottom). Finally, two diffusion matrices $D_{1}={\left\{D_{1}\left(i,j\right)\right\}}_{i=1,j=1}^{H,W}$ and $D_{2}={\left\{D_{2}\left(i,j\right)\right\}}_{i=1,j=1}^{H,W}$ with a size of H×W are obtained, which are used for forward-diffusion and backward-diffusion, respectively.

(4)Encryption process:

The gray image *P* with a size of H×W can be represented by a positive integer matrix with *H* rows and *W* columns, and its value range is [0,255]. According to [39], the steps of the encryption algorithm are as follows:

**Step 1.** Row-column permutation:

For the permutation operation, the row permutation operation is carried out according to the previously generated sequences *X* and *R*, and the column permutation operation is carried out according to the sequences *Y* and *T*.

For the row permutation operation, from i=1 to *N*, when x(i)≥0, the r(i)-th row of the image matrix *P* is inserted into the (H−r(i)+1)-th row, and the (r(i)+1)-th to the (H−r(i)+1)-th rows are moved up one row as a whole; when x(i)<0, the (H−r(i)+1)-th row of the image matrix *P* is inserted into the r(i)-th row, and the r(i)-th to the (H−r(i))-th rows are moved down one row as a whole. The row permutation operation is repeated *N* times, and finally, the matrix *B* is obtained after row permutation.

For the column permutation operation, from i=1 to *N*, when y(i)≥0, the t(i)-th column of the image matrix *B* is inserted into the (W−t(i)+1)-th column, and the (t(i)+1)-th to the (W−t(i)+1)-th columns are shifted to the left as a whole; when y(i)<0, the (W−t(i)+1)-th column of the image matrix *B* is inserted into the t(i)-th column, and the t(i)-th to the (W−t(i))-th columns are shifted to the right as a whole. The column permutation operation is repeated *N* times, and finally, the matrix *I* is obtained after column permutation.

**Step 2.** Forward-diffusion and backward-diffusion:

For the diffusion operation, the forward-diffusion is carried out according to the generated diffusion key $D_{1}$, and $S={\left\{S\left(i,j\right)\right\}}_{i=1,j=1}^{H,W}$ is the matrix after the forward-diffusion. According to the generated diffusion key $D_{2}$, the backward-diffusion is carried out, and $C={\left\{C\left(i,j\right)\right\}}_{i=1,j=1}^{H,W}$ is the matrix after the backward-diffusion.

For the forward-diffusion, the following diffusion methods are adopted:(4)$$S\left(i,j\right)=\left\{\begin{array}{cc} \mathrm{mod}(I\left(1,1\right)+D_{1}\left(1,1\right),256) & \mathrm{if}\;i=1,j=1,\\ \mathrm{mod}(I\left(1,j\right)+D_{1}\left(1,j\right)+S\left(1,j-1\right),256) & \mathrm{if}\;i=1,1<j\leq W,\\ \mathrm{mod}(I\left(i,1\right)+D_{1}\left(i,1\right)+S\left(i-1,1\right),256) & \mathrm{if}\;1<i\leq H,j=1,\\ \mathrm{mod}(I\left(i,j\right)+D_{1}\left(i,j\right)+S\left(i,j-1\right)+S\left(i-1,j\right),256) & \mathrm{if}\;1<i\leq H,1<j\leq W, \end{array}\right.$$
where i=1,2,⋯,H and j=1,2,⋯,W.

For the backward-diffusion, the following diffusion methods are adopted:(5)$$C\left(i,j\right)=\left\{\begin{array}{cc} \mathrm{mod}(S\left(H,W\right)+D_{2}\left(H,W\right),256) & \mathrm{if}\;i=H,j=W,\\ \mathrm{mod}(S\left(H,j\right)+D_{2}\left(H,j\right)+C\left(H,j+1\right),256) & \mathrm{if}\;i=H,1\leq j<W,\\ \mathrm{mod}(S\left(i,W\right)+D_{2}\left(i,W\right)+C\left(i+1,W\right),256) & \mathrm{if}\;1\leq i<H,j=W,\\ \mathrm{mod}(S\left(i,j\right)+D_{2}\left(i,j\right)+C\left(i,j+1\right)+C\left(i+1,j\right),256) & \mathrm{if}\;1\leq i<H,1\leq j<W, \end{array}\right.$$
where i=H,H−1,⋯,1 and j=W,W−1,⋯,1.

## 3. Security Analysis of the Original Encryption Algorithm

### 3.1. Overall Analysis

In cryptanalysis, the entire encryption algorithm can be cracked if the initial key parameters or equivalent key are obtained. Generally, the difficulty of cracking the initial key parameters is greater than that of cracking the equivalent key. If only the equivalent key can be solved, there is no need to crack the initial key parameters, because this will not only increase the difficulty and workload of cracking, but may even fail to crack. For the open-loop system, the generation of the encryption sequence is independent of the plaintext, that is the equivalent key is independent of the plaintext. It only needs to crack the equivalent key, and the cracking is relatively simple; for the closed-loop system, the generation of the encryption sequence is related to the plaintext, that is the equivalent key is related to the plaintext, so that the encryption sequence corresponding to different plaintexts is different, and the plaintext is not known in advance, so cracking the equivalent key will lose its generality. In this case, only the initial key parameters can be cracked.

According to the block diagram of the encryption algorithm shown in Figure 1, the initial key parameters of the chaotic map are independent of the plaintext image. When the initial key parameters remain unchanged and different plaintext images are encrypted, the permutation sequences *R* and *T* used for the two rounds of row–column permutation and the two diffusion matrices $D_{1}$ and $D_{2}$ are all unchanged, so it is easy to find the equivalent permutation key and equivalent diffusion key of the original algorithm by the chosen plaintext attack, thus cracking the entire encryption algorithm, that is to say, by selecting some plaintext information $P_{1},P_{2},\cdots ,P_{N}$ and the corresponding ciphertext information $C_{1},C_{2},\cdots ,C_{N}$ that is beneficial to the cracking, then deducing the secret key or equivalent key (represented by the symbol *K*) according to the encryption algorithm $E_{K}$ through these plaintext–ciphertext pairs, or finding an algorithm to derive $P_{N+1}$ from $C_{N+1}=E_{K}\left(P_{N+1}\right)$.

The whole encryption process can be equivalent to a round of global permutation and a round of global diffusion through the equivalent permutation key and the equivalent diffusion key. As shown in Figure 2, *P* is the plaintext gray image, *C* is the final ciphertext image, *I* is the image after global permutation, $K_{s}$ is the equivalent permutation key, and $K_{d}$ is the equivalent diffusion key.

After the above analysis, the original encryption algorithm has some defects in its structure. As shown in Figure 2, the forward-diffusion process is preprocessed by iterative optimization, and the iterative results are substituted into the backward-diffusion process to find the relationship between the two rounds of diffusion, so the equivalent diffusion key stream can be obtained by adjusting the individual pixel values of the chosen plaintext to crack the entire diffusion process. Then, on the basis of this analysis, the chosen plaintext attack is also used to crack the two rounds of the permutation process. The whole encryption algorithm can be finally cracked by carrying out cryptanalysis on each process of encryption.

### 3.2. Analysis of the Two-Round Diffusion Process of the Original Encryption Algorithm

#### 3.2.1. Pre-Analysis of Diffusion Process

According to the above overall analysis, the key to cracking the two-round diffusion process of the original encryption algorithm is to find the equivalent diffusion key and make it equivalent to one round of global diffusion. Therefore, this section mainly pre-analyzes the diffusion process. According to Equation (Equation 4), when 1<i≤H and i<j≤W, the pixel value S(i,j) is related to S(i,j−1) and S(i−1,j). Considering the characteristics of this kind of matrix, the following propositions are given.

**Proposition 1.** 
*For the matrix $A={\left\{A\left(k,l\right)\right\}}_{k=1,l=1}^{i,j}$ of size i×j, it satisfies that the values in the i-th row or j-th column are all 1, and the values of other positions are equal to the sum of the values in the position below and the position right, as shown in Figure 3a, that is*

$$A\left(k,l\right)=\left\{\begin{array}{cc} 1 & \mathrm{if}\;k=i,1\leq l\leq j\;\mathrm{or}\;1\leq k<i,l=1,\\ A(k,l-1)+A(k-1,l) & \mathrm{if}\;1\leq k\leq i-1,1\leq l\leq j-1, \end{array}\right.$$

*where k and l represent the row number and column number in the matrix A, respectively. Then, the element value A(k,l) in the k-th row and l-th column of this matrix can be expressed as $C_{i-k+j-l}^{j-l}$, namely*

$$A\left(k,l\right)=C_{i-k+j-l}^{j-l}=\frac{\left(i-k+j-l)(i-k+j-l-1)\cdots (i-k+1\right)}{(j-l)!}.$$

*Since the distribution of element values in matrix A is symmetric, the element value A(k,l) in the k-th row and l-th column of the matrix is equal to the element value A(l,k) in the l-th row and k-th column of the matrix, namely*

$$A\left(k,l\right)=A\left(l,k\right)=C_{i-k+j-l}^{i-k}=\frac{\left(i-k+j-l)(i-k+j-l-1)\cdots (j-l+1\right)}{(i-k)!}.$$



**Proof.** According to the definition of Pascal’s triangle, we can obtain the structure of Figure 3b. Matrix *A* is now rotated by $135^{\circ }$ counterclockwise, which coincides with part of Figure 3b. At this time, the properties of Pascal’s triangle can be used to obtain the values of each position of the matrix. The following are some basic properties of Pascal’s triangle:(1) Each number in Pascal’s triangle is equal to the sum of its left and right numbers in the previous line.(2) The *q*-th number in row *p* of Pascal’s triangle is equal to the (p−q+1)-th number in row *p*.(3) The *q*-th number in row *p* of Pascal’s triangle is $C_{p-1}^{q-1}$, where $C_{p-1}^{q-1}$ represents the operations of permutation and combination in mathematics, namely
(6)$$C_{p-1}^{q-1}=\frac{\left(p-1)(p-2)\cdots (p-q+1\right)}{(q-1)!}.$$The positions referred to the above properties all correspond to the positions in Pascal’s triangle. Therefore, in order to obtain each value in matrix *A*, the corresponding relationship between the values in matrix *A* and Pascal’s triangle are required. Figure 4 shows the corresponding relationship between the values in matrix *A* and Pascal’s triangle. The left side represents the position of each element in matrix *A*, and the right side corresponds to the row number of Pascal’s triangle in Figure 4. Because the row and column descriptions of matrix *A* and Pascal’s triangle are different in the analysis process, therefore, to show the difference, a(∗,∗) is used to represent the element values in Pascal’s triangle and A(∗,∗) is used to represent the element values of matrix *A*.Analyzing the corresponding relationship in Figure 4, it is seen that the row number of Pascal’s triangle corresponds to the sum of the row number and column number of the elements in matrix *A*. This sum is a fixed value. The corresponding fixed value of the first row of Pascal’s triangle is (i+j), and then, as the row increases, the sum value will decrease gradually with a tolerance of 1, so the element at position (k,l) in matrix *A* is in row *p*(p=i−k+j−l+1) of Pascal’s triangle. The first element of the *p*-th row of Pascal’s triangle is a(i−p+1,j); the row number of the element *a* increases with a tolerance of 1, and the column number decreases with a tolerance of 1, Therefore, the element in the position (k,l) of the matrix *A* corresponds to the *q*-th (q=j−l+1) position of Pascal’s triangle. It can also be seen from the symmetry of Pascal’s triangle that A(k,l) is also equal to the value of the (i−k+1)-th position of the *k*-th row.According to the one-to-one correspondence between the two positions in the above analysis and Property (3) of Pascal’s triangle, substitute p=i−k+j−l+1, q=j−l+1 or q=i−k+1 into Equation (Equation 6) to obtain the value of position (k,l) in matrix *A* as $C_{i-k+j-l}^{j-l}$ or $C_{i-k+j-l}^{i-k}$, namely
$$A\left(k,l\right)=C_{i-k+j-l}^{j-l}=\frac{\left(i-k+j-l)(i-k+j-l-1)\cdots (i-k+1\right)}{(j-l)!},$$
or
$$A\left(k,l\right)=A\left(l,k\right)=C_{i-k+j-l}^{i-k}=\frac{\left(i-k+j-l)(i-k+j-l-1)\cdots (j-l+1\right)}{(i-k)!}.$$
Proposition 1 is proven.    □

**Proposition 2.** 
*In the original encryption algorithm, the two-round diffusion encryption represented by the forward-diffusion encryption Equation (Equation 4) and the backward-diffusion encryption Equation (Equation 5) can be equivalent to one-round global diffusion encryption, namely*

$$C\left(i,j\right)=\left\{\begin{array}{cc} \mathrm{mod}(f_{1}\left(I,H,W\right)+K_{d}\left(H,W\right),256) & \mathrm{if}\;i=H,j=W,\\ \mathrm{mod}(f_{1}\left(I,H,j\right)+K_{d}\left(H,j\right)+C\left(H,j+1\right),256) & \mathrm{if}\;i=H,1\leq j<W,\\ \mathrm{mod}(f_{1}\left(I,i,W\right)+K_{d}\left(i,W\right)+C\left(i+1,W\right),256) & \mathrm{if}\;1\leq i<H,j=W,\\ \mathrm{mod}(f_{1}\left(I,i,j\right)+K_{d}\left(i,j\right)+C\left(i,j+1\right)+C\left(i+1,j\right),256) & \mathrm{if}\;1\leq i<H,1\leq j<W, \end{array}\right.$$

*where $f_{1}\left(I,i,j\right)$ represents the value related to plaintext image I, and its specific expression is as follows:*

$$f_{1}\left(I,i,j\right)=\left\{\begin{array}{cc} I(1,1) & \mathrm{if}\;i=1,j=1,\\ \sum_{l=1}^{j}I\left(1,l\right) & \mathrm{if}\;i=1,j=2,3,\cdots ,W,\\ \sum_{k=1}^{i}I\left(k,1\right) & \mathrm{if}\;i=2,3,\cdots ,H,j=1,\\ \sum_{k=1}^{i}\sum_{l=1}^{j}C_{i-k+j-l}^{i-k}I\left(k,l\right) & \mathrm{if}\;i=2,3,\cdots ,H,j=2,3,\cdots ,W, \end{array}\right.$$

*then $K_{d}={\left\{K_{d}\left(i,j\right)\right\}}_{i=1,j=1}^{H,W}$ is the equivalent diffusion key, $K_{d}=f_{2}\left(D_{1},i,j\right)+D_{2}\left(i,j\right)$ represents the value related to the forward-diffusion key $D_{1}$ and the backward-diffusion key $D_{2}$, and*

$$f_{2}\left(D_{1},i,j\right)=\left\{\begin{array}{cc} D_{1}\left(1,1\right) & \mathrm{if}\;i=1,j=1,\\ \sum_{l=1}^{j}D_{1}\left(1,l\right) & \mathrm{if}\;i=1,j=2,3,\cdots ,W,\\ \sum_{k=1}^{i}D_{1}\left(k,1\right) & \mathrm{if}\;i=2,3,\cdots ,H,j=1,\\ \sum_{k=1}^{i}\sum_{l=1}^{j}C_{i-k+j-l}^{i-k}D_{1}\left(k,l\right) & if\;i=2,3,\cdots ,H,j=2,3,\cdots ,W, \end{array}\right.$$



**Proof.** For the forward-diffusion process, Equation (Equation 4) is preprocessed step by step and iteratively:(1) When i=1, j=1, there is
(7)$$\begin{array}{c} S\left(1,1\right)=\mathrm{mod}(I\left(1,1\right)+D_{1}\left(1,1\right),256). \end{array}$$(2) When i=1, j=2,3,⋯,W, there is
(8)$$\begin{array}{cc} S(1,j) & =\mathrm{mod}(I\left(1,j\right)+D_{1}\left(1,j\right)+S\left(1,j-1\right),256)\\ \; & =\mathrm{mod}(I\left(1,j\right)+D_{1}\left(1,j\right)+I\left(1,j-1\right)+D_{1}\left(1,j-1\right)+S\left(1,j-2\right),256)\\ \; & \vdots \\ \; & =\mathrm{mod}(I\left(1,j\right)+D_{1}\left(1,j\right)+I\left(1,j-1\right)+D_{1}\left(1,j-1\right)+\cdots +I\left(1,1\right)+D_{1}\left(1,1\right),256)\\ \; & =\mathrm{mod}(\sum_{l=1}^{j}I\left(1,l\right)+\sum_{l=1}^{j}D_{1}\left(1,l\right),256). \end{array}$$(3) When i=2,3,⋯,H, j=1, there is
(9)$$\begin{array}{cc} S(i,1) & =\mathrm{mod}(I\left(i,1\right)+D_{1}\left(i,1\right)+S\left(i-1,1\right),256)\\ \; & =\mathrm{mod}(I\left(i,1\right)+D_{1}\left(i,1\right)+I\left(i-1,1\right)+D_{1}\left(i-1,1\right)+S\left(i-2,1\right),256)\\ \; & \vdots \\ \; & =\mathrm{mod}(I\left(i,1\right)+D_{1}\left(i,1\right)+I\left(i-1,1\right)+D_{1}\left(i-1,1\right)+\cdots +I\left(1,1\right)+D_{1}\left(1,1\right),256)\\ \; & =\mathrm{mod}(\sum_{k=1}^{i}I\left(k,1\right)+\sum_{k=1}^{i}D_{1}\left(k,1\right),256). \end{array}$$(4) When i=2,3,⋯,H, j=2,3,⋯,W, there is
$$\begin{array}{c} S\left(i,j\right)=\mathrm{mod}(I\left(i,j\right)+D_{1}\left(i,j\right)+S\left(i,j-1\right)+S\left(i-1,j\right),256). \end{array}$$After substituting the expressions of S(i,j−1) and S(i−1,j), there is
$$\begin{array}{c} \begin{array}{cc} S(i,j) & =\mathrm{mod}(I\left(i,j\right)+D_{1}\left(i,j\right)+\\ & \;I\left(i,j-1\right)+D_{1}\left(i,j-1\right)+S\left(i,j-2\right)+S\left(i-1,j-1\right)+\\ & \;I\left(i-1,j\right)+D_{1}\left(i-1,j\right)+S\left(i-2,j\right)+S\left(i-1,j-1\right),256). \end{array} \end{array}$$Arrange the pixel values in the same matrix into the same row:
$$\begin{array}{c} \begin{array}{cc} S(i,j) & =\mathrm{mod}(I(i,j)+I(i,j-1)+I(i-1,j)+\\ & \;D_{1}\left(i,j\right)+D_{1}\left(i,j-1\right)+D_{1}\left(i-1,j\right)+\\ & \;S(i,j-2)+S(i-2,j)+2S(i-1,j-1),256). \end{array} \end{array}$$The constant coefficients corresponding to the pixel values of each position obviously meet the properties in Proposition 1, and the coefficients are replaced by the elements of Pascal’s triangle:
$$\begin{array}{c} \begin{array}{cc} S(i,j) & =\mathrm{mod}(C_{0}^{0}I\left(i,j\right)+C_{1}^{1}I\left(i,j-1\right)+C_{1}^{0}I\left(i-1,j\right)+\\ & \;C_{0}^{0}D_{1}\left(i,j\right)+C_{1}^{1}D_{1}\left(i,j-1\right)+C_{1}^{0}D_{1}\left(i-1,j\right)+\\ & \;S(i,j-2)+S(i-2,j)+2S(i-1,j-1),256). \end{array} \end{array}$$By analyzing the above iterative rule, the pixel value of matrix S(i,j) can be cyclically iterated to S(1,1). According to the forward-diffusion formula, it can be known that $S\left(1,1\right)=\mathrm{mod}(I\left(1,1\right)+D_{1}\left(1,1\right),256)$, so the pixel value of matrix *S* can be eliminated in the end, and the iteration is as follows:
$$\begin{array}{c} \begin{array}{cc} S(i,j) & =\mathrm{mod}(C_{0}^{0}I\left(i,j\right)+C_{1}^{1}I\left(i,j-1\right)+C_{1}^{0}I\left(i-1,j\right)+\cdots +C_{i-k+j-l}^{i-k}I\left(k,l\right)+\cdots +C_{i+j-2}^{i-1}I\left(1,1\right)+\\ & \;C_{0}^{0}D_{1}\left(i,j\right)+C_{1}^{1}D_{1}\left(i,j-1\right)+C_{1}^{0}D_{1}\left(i-1,j\right)+\cdots +C_{i-k+j-l}^{i-k}D_{1}\left(k,l\right)+\cdots +C_{i+j-2}^{i-1}D_{1}\left(1,1\right),256). \end{array} \end{array}$$At this point, the above equation can be summarized and simplified as follows:
(10)$$\begin{array}{c} S\left(i,j\right)=\mathrm{mod}(\sum_{k=1}^{i}\sum_{l=1}^{j}C_{i-k+j-l}^{i-k}I\left(k,l\right)+\sum_{k=1}^{i}\sum_{l=1}^{j}C_{i-k+j-l}^{i-k}D_{1}\left(k,l\right),256). \end{array}$$
where $C_{i-k+j-l}^{i-k}$ represents the constant coefficient of the pixel value.After the above preprocessing, the iterative result of the forward-diffusion process is determined by Equations (Equation 7)–(Equation 10), and the final iterative result is only associated with the plaintext image *I* and forward-diffusion key $D_{1}$. For the convenience of further derivation and processing, it is suggested to uniformly set them as
(11)$$\begin{array}{c} S\left(i,j\right)=\mathrm{mod}(f_{1}\left(I,i,j\right)+f_{2}\left(D_{1},i,j\right),256). \end{array}$$
where $f_{1}\left(I,i,j\right)$ and $f_{2}\left(D_{1},i,j\right)$, respectively, represent the value related to the intermediate ciphertext image *I* and the value related to forward-diffusion key $D_{1}$. Their specific expression is as follows:
(12)$$f_{1}\left(I,i,j\right)=\left\{\begin{array}{cc} I(1,1) & \mathrm{if}\;i=1,j=1,\\ \sum_{l=1}^{j}I\left(1,l\right) & \mathrm{if}\;i=1,j=2,3,\cdots ,W,\\ \sum_{k=1}^{i}I\left(k,1\right) & \mathrm{if}\;i=2,3,\cdots ,H,j=1,\\ \sum_{k=1}^{i}\sum_{l=1}^{j}C_{i-k+j-l}^{i-k}I\left(k,l\right) & \mathrm{if}\;i=2,3,\cdots ,H,j=2,3,\cdots ,W, \end{array}\right.$$
(13)$$f_{2}\left(D_{1},i,j\right)=\left\{\begin{array}{cc} D_{1}\left(1,1\right) & \mathrm{if}\;i=1,j=1,\\ \sum_{l=1}^{j}D_{1}\left(1,l\right) & \mathrm{if}\;i=1,j=2,3,\cdots ,W,\\ \sum_{k=1}^{i}D_{1}\left(k,1\right) & \mathrm{if}\;i=2,3,\cdots ,H,j=1,\\ \sum_{k=1}^{i}\sum_{l=1}^{j}C_{i-k+j-l}^{i-k}D_{1}\left(k,l\right) & \mathrm{if}\;i=2,3,\cdots ,H,j=2,3,\cdots ,W, \end{array}\right.$$In order to obtain the equivalent diffusion key, it is also necessary to substitute the forward-diffusion process into the backward-diffusion process, analyze the relationship between them, and convert the two-round diffusion to the one-round diffusion. By substituting Equation (Equation 11) into Equation (Equation 5), the following formula can be obtained as
(14)$$C\left(i,j\right)=\left\{\begin{array}{cc} \mathrm{mod}(f_{1}\left(I,H,W\right)+f_{2}\left(D_{1},H,W\right)+D_{2}\left(H,W\right),256) & \mathrm{if}\;i=H,j=W,\\ \mathrm{mod}(f_{1}\left(I,H,j\right)+f_{2}\left(D_{1},H,j\right)+D_{2}\left(H,j\right)+C\left(H,j+1\right),256) & \mathrm{if}\;i=H,1\leq j<W,\\ \mathrm{mod}(f_{1}\left(I,i,W\right)+f_{2}\left(D_{1},i,W\right)+D_{2}\left(i,W\right)+C\left(i+1,W\right),256) & \mathrm{if}\;1\leq i<H,j=W,\\ \mathrm{mod}(f_{1}\left(I,i,j\right)+f_{2}\left(D_{1},i,j\right)+D_{2}\left(i,j\right)+C\left(i,j+1\right)+C\left(i+1,j\right),256) & \mathrm{if}\;1\leq i<H,1\leq j<W, \end{array}\right.$$The analysis of the above formula shows that, in the iterative process, the intermediate value *S* was eliminated, and the entire global diffusion process is only associated with plaintext image *I*, ciphertext image *C*, and the two-round diffusion keys $D_{1}$ and $D_{2}$. Since $D_{1}$ and $D_{2}$ are constants, then $f_{2}\left(D_{1},i,j\right)+D_{2}\left(i,j\right)$ is a constant. The related parts of the two-round diffusion keys $D_{1}$ and $D_{2}$ can be regarded as a whole, and the equivalent diffusion key $K_{d}=f_{2}\left(D_{1},i,j\right)+D_{2}\left(i,j\right)$, then Equation (Equation 14) can be simplified as
(15)$$C\left(i,j\right)=\left\{\begin{array}{cc} \mathrm{mod}(f_{1}\left(I,H,W\right)+K_{d}\left(H,W\right),256) & \mathrm{if}\;i=H,j=W,\\ \mathrm{mod}(f_{1}\left(I,H,j\right)+K_{d}\left(H,j\right)+C\left(H,j+1\right),256) & \mathrm{if}\;i=H,1\leq j<W,\\ \mathrm{mod}(f_{1}\left(I,i,W\right)+K_{d}\left(i,W\right)+C\left(i+1,W\right),256) & \mathrm{if}\;1\leq i<H,j=W,\\ \mathrm{mod}(f_{1}\left(I,i,j\right)+K_{d}\left(i,j\right)+C\left(i,j+1\right)+C\left(i+1,j\right),256) & \mathrm{if}\;1\leq i<H,1\leq j<W, \end{array}\right.$$According to the above analysis, there is an equivalent diffusion key $K_{d}$, and Equation (Equation 15) clearly indicates that there is a fixed relationship between output *C* and input *I* through the equivalent diffusion key $K_{d}$. Therefore, the two-round diffusion encryption represented by forward-diffusion encryption and backward-diffusion encryption in the original encryption algorithm can be equivalent to the one-round global diffusion encryption.    □

#### 3.2.2. The Equivalent Diffusion Key

To further crack the diffusion process, the equivalent diffusion key $K_{d}$ of the diffusion process is reversely obtained according to Equation (Equation 15) as
(16)$$K_{d}\left(i,j\right)=\left\{\begin{array}{cc} \mathrm{mod}(C\left(H,W\right)-f_{1}\left(I,H,W\right),256)+256n & \mathrm{if}\;i=H,j=W,\\ \mathrm{mod}(C\left(H,j\right)-f_{1}\left(I,H,j\right)-C\left(H,j+1\right),256)+256n & \mathrm{if}\;i=H,1\leq j<W,\\ \mathrm{mod}(C\left(i,W\right)-f_{1}\left(I,i,W\right)-C\left(i+1,W\right),256)+256n & \mathrm{if}\;1\leq i<H,j=W,\\ \mathrm{mod}(C\left(i,j\right)-f_{1}\left(I,i,j\right)-C\left(i,j+1\right)-C\left(i+1,j\right),256)+256n & \mathrm{if}\;1\leq i<H,1\leq j<W, \end{array}\right.$$
where i=H,H−1,⋯,1, j=W,W−1,⋯,1, and n∈N.

Further analysis shows that the equivalent diffusion key $K_{d}$ is subsequently used for backward-diffusion to obtain the plaintext image, which requires modulo to 256, so the influence of 256n can be ignored in Equation (Equation 16), namely
(17)$$K_{d}\left(i,j\right)=\left\{\begin{array}{cc} \mathrm{mod}(C\left(H,W\right)-f_{1}\left(I,H,W\right),256) & \mathrm{if}\;i=H,j=W,\\ \mathrm{mod}(C\left(H,j\right)-f_{1}\left(I,H,j\right)-C\left(H,j+1\right),256) & \mathrm{if}\;i=H,1\leq j<W,\\ \mathrm{mod}(C\left(i,W\right)-f_{1}\left(I,i,W\right)-C\left(i+1,W\right),256) & \mathrm{if}\;1\leq i<H,j=W,\\ \mathrm{mod}(C\left(i,j\right)-f_{1}\left(I,i,j\right)-C\left(i,j+1\right)-C\left(i+1,j\right),256) & \mathrm{if}\;1\leq i<H,1\leq j<W, \end{array}\right.$$
where i=H,H−1,⋯,1 and j=W,W−1,⋯,1.

According to Equation (Equation 17), to obtain the equivalent diffusion key $K_{d}$, select an all-zero plaintext image $P_{0}={\left\{P_{0}\left(i,j\right)\right\}}_{i=1,j=1}^{H,W}=0$ according to the chosen plaintext attack, then encrypt to the corresponding ciphertext image $C_{0}={\left\{C_{0}\left(i,j\right)\right\}}_{i=1,j=1}^{H,W}$, because the all-zero plaintext is not influenced by the permutation operation; after permutation encryption, $I_{0}$ is still an all-zero matrix $I_{0}={\left\{I_{0}\left(i,j\right)\right\}}_{i=1,j=1}^{H,W}=0$. From Equation (Equation 12), we can know that $f_{1}\left(I_{0},i,j\right)=0$ (i=1,2,⋯,H, j=1,2,⋯,W) and $C_{0}$ as a known condition, and then, substitute them into Equation (Equation 17) to find the equivalent diffusion key as
(18)$$K_{d}\left(i,j\right)=\left\{\begin{array}{cc} \mathrm{mod}(C_{0}\left(H,W\right),256) & \mathrm{if}\;i=H,j=W,\\ \mathrm{mod}(C_{0}\left(H,j\right)-C_{0}\left(H,j+1\right),256) & \mathrm{if}\;i=H,1\leq j<W,\\ \mathrm{mod}(C_{0}\left(i,W\right)-C_{0}\left(i+1,W\right),256) & \mathrm{if}\;1\leq i<H,j=W,\\ \mathrm{mod}(C_{0}\left(i,j\right)-C_{0}\left(i,j+1\right)-C_{0}\left(i+1,j\right),256) & \mathrm{if}\;1\leq i<H,1\leq j<W, \end{array}\right.$$
where i=H,H−1,⋯,1 and j=W,W−1,⋯,1.

According to the above analysis, the algorithm to crack the equivalent diffusion key $K_{d}$ is shown in Algorithm 1.
**Algorithm 1** The procedure of cracking the equivalent diffusion key $K_{d}$.**Input:** 
$P_{0}$, $C_{0}$**Output:** 
$K_{d}$1:$\left[H,W\right]=\mathrm{size}\left(P_{0}\right)$2:$C_{0}\leftarrow \mathrm{Encrypt}\left(P_{0}\right)$; $K_{d}\leftarrow \mathrm{zeros}\left(H,W\right)$3:$K_{d}\left(H,W\right)\leftarrow \mathrm{mod}\left(C_{0}\left(H,W\right),256\right)$4:**for** j=W−1 to 1 **do**5:   $K_{d}\left(H,j\right)\leftarrow \mathrm{mod}\left(C_{0}\left(H,j\right)-C_{0}\left(H,j+1\right),256\right)$6:**end for**7:**for** i=H−1 to 1 **do**8:   $K_{d}\left(i,W\right)\leftarrow \mathrm{mod}\left(C_{0}\left(i,W\right)-C_{0}\left(i+1,W\right),256\right)$9:**end for**10:**for** i=H−1 to 1 **do**11:   **for** j=W−1 to 1 **do**12:     $K_{d}\left(i,j\right)\leftarrow \mathrm{mod}\left(C_{0}\left(i,j\right)-C_{0}\left(i,j+1\right)-C_{0}\left(i+1,j\right),256\right)$13:   **end for**14:**end for**15:**return** $K_{d}$

#### 3.2.3. Cracking Diffusion Encryption Process Using Equivalent Diffusion Key

Based on the equivalent diffusion key $K_{d}$, backward-diffusion is carried out according to Equation (Equation 15) and the intermediate ciphertext image before diffusion is reversely calculated. The process of backward-diffusion is as follows:(19)$$f_{1}\left(I,i,j\right)=\left\{\begin{array}{cc} \mathrm{mod}(C\left(H,W\right)-K_{d}\left(H,W\right),256)+256n & \mathrm{if}\;i=H,j=W,\\ \mathrm{mod}(C\left(H,j\right)-K_{d}\left(H,j\right)-C\left(H,j+1\right),256)+256n & \mathrm{if}\;i=H,1\leq j<W,\\ \mathrm{mod}(C\left(i,W\right)-K_{d}\left(i,W\right)-C\left(i+1,W\right),256)+256n & \mathrm{if}\;1\leq i<H,j=W,\\ \mathrm{mod}(C\left(i,j\right)-K_{d}\left(i,j\right)-C\left(i,j+1\right)-C\left(i+1,j\right),256)+256n & \mathrm{if}\;1\leq i<H,1\leq j<W, \end{array}\right.$$
where i=H,H−1,⋯,1, j=W,W−1,⋯,1, and n∈N.

Further analysis showed that, in order to offset the influence of 256n in Equation (Equation 19), the modulo method of 256 was also adopted to obtain the single pixel of the intermediate ciphertext image *I*. Therefore, Equation (Equation 19) can be simplified as follows:(20)$$f_{1}\left(I,i,j\right)=\left\{\begin{array}{cc} \mathrm{mod}(C\left(H,W\right)-K_{d}\left(H,W\right),256) & \mathrm{if}\;i=H,j=W,\\ \mathrm{mod}(C\left(H,j\right)-K_{d}\left(H,j\right)-C\left(H,j+1\right),256) & \mathrm{if}\;i=H,1\leq j<W,\\ \mathrm{mod}(C\left(i,W\right)-K_{d}\left(i,W\right)-C\left(i+1,W\right),256) & \mathrm{if}\;1\leq i<H,j=W,\\ \mathrm{mod}(C\left(i,j\right)-K_{d}\left(i,j\right)-C\left(i,j+1\right)-C\left(i+1,j\right),256) & \mathrm{if}\;1\leq i<H,1\leq j<W, \end{array}\right.$$
where i=H,H−1,⋯,1 and j=W,W−1,⋯,1.

According to Equation (Equation 20), the sum of pixels $f_{1}\left(I,i,j\right)$ related to the intermediate ciphertext image *I* before diffusion can be obtained by using the equivalent diffusion key $K_{d}$ for backward-diffusion. To recover the intermediate ciphertext image *I* before diffusion, the specific pixel value I(i,j) of each position of the image needs to be obtained. According to the previous analysis process, the relationship between the sum of pixels $f_{1}\left(I,i,j\right)$ related to the intermediate ciphertext and the pixel value I(i,j) of each position is determined by Equation (Equation 12). Now, Equation (Equation 12) is used to obtain the pixel value I(i,j) of the intermediate ciphertext. The process is as follows:

(1) When i=1, j=1, there is
(21)$$\begin{array}{cc} I(1,1) & =f_{1}\left(I,1,1\right)\\ \; & =\mathrm{mod}(C\left(1,1\right)-K_{d}\left(1,1\right)-C\left(1,2\right)-C\left(2,1\right),256). \end{array}$$

(2) When i=1, j=2,3,⋯,W, there is
(22)$$\begin{array}{cc} I(1,j) & =\sum_{l=1}^{j}I\left(1,l\right)-\sum_{l=1}^{j-1}I\left(1,l\right)\\ \; & =f_{1}\left(I,1,j\right)-f_{1}\left(I,1,j-1\right)\\ \; & =\left\{\begin{array}{cc} f_{1}\left(I,1,W\right)-f_{1}\left(I,1,W-1\right) & \mathrm{if}\;j=W,\\ f_{1}\left(I,1,j\right)-f_{1}\left(I,1,j-1\right) & \mathrm{if}\;1<j<W. \end{array}\right. \end{array}$$

That is, when j=W,
$$\begin{array}{c} \begin{array}{cc} I(1,W) & =\mathrm{mod}(C\left(1,W\right)-K_{d}\left(1,W\right)-C\left(2,W\right)-\\ & \;(C\left(1,W-1\right)-K_{d}\left(1,W-1\right)-C\left(1,W\right)-C\left(2,W-1\right)),256). \end{array} \end{array}$$

When 1<j<W,
$$\begin{array}{c} \begin{array}{cc} I(1,j) & =\mathrm{mod}(C\left(1,j\right)-K_{d}\left(1,j\right)-C\left(1,j+1\right)-C\left(2,j\right)-\\ & \;(C\left(1,j-1\right)-K_{d}\left(1,j-1\right)-C\left(1,j\right)-C\left(2,j-1\right)),256). \end{array} \end{array}$$

(3) When i=2,3,⋯,H, j=1, there is
(23)$$\begin{array}{cc} I(i,1) & =\sum_{k=1}^{i}I\left(k,1\right)-\sum_{k=1}^{i-1}I\left(k,1\right)\\ \; & =f_{1}\left(I,i,1\right)-f_{1}\left(I,i-1,1\right)\\ \; & =\left\{\begin{array}{cc} f_{1}\left(I,H,1\right)-f_{1}\left(I,H-1,1\right) & \mathrm{if}\;i=H,\\ f_{1}\left(I,i,1\right)-f_{1}\left(I,i-1,1\right) & \mathrm{if}\;1<i<H. \end{array}\right. \end{array}$$

That is, when i=H,
$$\begin{array}{c} \begin{array}{cc} I(H,1) & =\mathrm{mod}(C\left(H,1\right)-K_{d}\left(H,1\right)-C\left(H,2\right)-\\ & \;(C\left(H-1,1\right)-K_{d}\left(H-1,1\right)-C\left(H,1\right)-C\left(H-1,2\right)),256). \end{array} \end{array}$$

When 1<i<H,
$$\begin{array}{c} \begin{array}{cc} I(i,1) & =\mathrm{mod}(C\left(i,1\right)-K_{d}\left(i,1\right)-C\left(i,2\right)-C\left(i+1,1\right)-\\ & \;(C\left(i-1,1\right)-K_{d}\left(i-1,1\right)-C\left(i-1,2\right)-C\left(i,1\right)),256). \end{array} \end{array}$$

(4) When i=2,3,⋯,H, j=2,3,⋯,W, it can be seen from Property (1) of Pascal’s triangle that
(24)$$\begin{array}{cc} I(i,j) & =\sum_{k=1}^{i}\sum_{l=1}^{j}C_{i-k+j-l}^{i-k}I\left(k,l\right)-\sum_{k=1}^{i-1}\sum_{l=1}^{j}C_{i-k+j-l}^{i-k}I\left(k,l\right)-\sum_{k=1}^{i}\sum_{l=1}^{j-1}C_{i-k+j-l}^{i-k}I\left(k,l\right)\\ \; & =f_{1}\left(I,i,j\right)-f_{1}\left(I,i-1,j\right)-f_{1}\left(I,i,j-1\right)\\ \; & =\left\{\begin{array}{cc} f_{1}\left(I,H,W\right)-f_{1}\left(I,H-1,W\right)-f_{1}\left(I,H,W-1\right) & \mathrm{if}\;i=H,j=W\\ f_{1}\left(I,H,j\right)-f_{1}\left(I,H-1,j\right)-f_{1}\left(I,H,j-1\right) & \mathrm{if}\;i=H,1<j<W\\ f_{1}\left(I,i,W\right)-f_{1}\left(I,i-1,W\right)-f_{1}\left(I,i,W-1\right) & \mathrm{if}\;1<i<H,j=W\\ f_{1}\left(I,i,j\right)-f_{1}\left(I,i-1,j\right)-f_{1}\left(I,i,j-1\right) & \mathrm{if}\;1<i<H,1<j<W. \end{array}\right. \end{array}$$

That is, when i=H, j=W,
$$\begin{array}{c} \begin{array}{cc} I(H,W) & =\mathrm{mod}(C\left(H,W\right)-K_{d}\left(H,W\right)-\\ & \;(C\left(H-1,W\right)-K_{d}\left(H-1,W\right)-C\left(H,W\right))-\\ & \;(C\left(H,W-1\right)-K_{d}\left(H,W-1\right)-C\left(H,W\right)),256). \end{array} \end{array}$$

When i=H, 1<j<W,
$$\begin{array}{c} \begin{array}{cc} I(H,j) & =\mathrm{mod}(C\left(H,j\right)-K_{d}\left(H,j\right)-C\left(H,j+1\right)-\\ & \;(C\left(H-1,j\right)-K_{d}\left(H-1,j\right)-C\left(H-1,j+1\right)-C\left(H,j\right))-\\ & \;(C\left(H,j-1\right)-K_{d}\left(H,j-1\right)-C\left(H,j\right)),256). \end{array} \end{array}$$

When 1<i<H, j=W,
$$\begin{array}{c} \begin{array}{cc} I(i,W) & =\mathrm{mod}(C\left(i,W\right)-K_{d}\left(i,W\right)-C\left(i+1,W\right)-\\ & \;(C\left(i-1,W\right)-K_{d}\left(i-1,W\right)-C\left(i,W\right))-\\ & \;(C\left(i,W-1\right)-K_{d}\left(i,W-1\right)-C\left(i,W\right)-C\left(i+1,W\right)),256). \end{array} \end{array}$$

When 1<i<H, 1<j<W,
$$\begin{array}{c} \begin{array}{cc} I(i,j) & =\mathrm{mod}(C\left(i,j\right)-K_{d}\left(i,j\right)-C\left(i,j+1\right)-C\left(i+1,j\right)-\\ & \;(C\left(i-1,j\right)-K_{d}\left(i-1,j\right)-C\left(i-1,j+1\right)-C\left(i,j\right))-\\ & \;(C\left(i,j-1\right)-K_{d}\left(i,j-1\right)-C\left(i,j\right)-C\left(i+1,j-1\right)),256). \end{array} \end{array}$$

The pixel values I(i,j) of each position of the intermediate ciphertext image *I* can be determined according to Equations (Equation 21)–(Equation 24). However, when the sum of pixels $f_{1}\left(I,i,j\right)$ related to the intermediate ciphertext is obtained according to Equation (Equation 20), the influence of 256n is ignored for the convenience of calculation and derivation. Therefore, when restoring the pixel values at each position of the intermediate ciphertext, the modulo of 256 can be taken to offset the influence, and the range of each pixel value is guaranteed to be [0,255]. When the value of *P* is determined, then *I* will be determined, and we can set the matrix $F={\left\{F\left(i,j\right)\right\}}_{i=1,j=1}^{H,W}$, where $F\left(i,j\right)=f_{1}\left(I,i,j\right)$. Therefore, the formula for calculating the pixel value I(i,j) of each position of the intermediate ciphertext image can be optimized as follows:(25)$$I\left(i,j\right)=\left\{\begin{array}{cc} \mathrm{mod}(F(1,1),256) & \mathrm{if}\;i=1,j=1,\\ \mathrm{mod}(F(1,j)-F(1,j-1),256) & \mathrm{if}\;i=1,1<j\leq W,\\ \mathrm{mod}(F(i,1)-F(i-1,1),256) & \mathrm{if}\;1<i\leq H,j=1,\\ \mathrm{mod}(F(i,j)-F(i-1,j)-F(i,j-1),256) & \mathrm{if}\;1<i\leq H,1<j\leq W, \end{array}\right.$$
where i=H,H−1,⋯,1, j=W,W−1,⋯,1, then without the help of the sum of pixels $f_{1}\left(I,i,j\right)$ related to the intermediate ciphertext, the pixel values of each position of the intermediate ciphertext *I* can be calculated according to the direct relationship F(i,j) between the intermediate ciphertext image *I* and the ciphertext image *C* analyzed above.

According to the above analysis process, the equivalent diffusion key $K_{d}$ is used to crack the diffusion encryption process, as shown in Algorithm 2.
**Algorithm 2** The procedure of cracking the diffusion process.**Input:** 
*C*, $K_{d}$**Output:** 
*I*1:[H,W]=size(C)2:F←zeros(H,W); I←zeros(H,W)3:$F\left(H,W\right)\leftarrow \mathrm{mod}(C\left(H,W\right)-K_{d}\left(H,W\right),256)$4:**for** j=W−1 to 1 **do**5:   $F\left(H,j\right)\leftarrow \mathrm{mod}(C\left(H,j\right)-K_{d}\left(H,j\right)-C\left(H,j+1\right),256)$6:**end for**7:**for** i=H−1 to 1 **do**8:   $F\left(i,W\right)\leftarrow \mathrm{mod}(C\left(i,W\right)-K_{d}\left(i,W\right)-C\left(i+1,W\right),256)$9:**end for**10:**for** i=H−1 to 1 **do**11:   **for** j=W−1 to 1 **do**12:     $F\left(i,j\right)\leftarrow \mathrm{mod}(C\left(i,j\right)-K_{d}\left(i,j\right)-C\left(i,j+1\right)-C\left(i+1,j\right),256)$13:   **end for**14:**end for**15:I(1,1)←mod(F(1,1),256)16:**for** j=2 to *W* **do**17:   I(1,j)←mod(F(1,j)−F(1,j−1),256)18:**end for**19:**for** i=2 to *H* **do**20:   I(i,1)←mod(F(i,1)−F(i−1,1),256)21:**end for**22:**for** i=2 to *H* **do**23:   **for** j=2 to *W* **do**24:     I(i,j)←mod(F(i,j)−F(i−1,j)−F(i,j−1),256)25:   **end for**26:**end for**27:**return** *I*

### 3.3. Cracking the Permutation Process

On the basis of cracking the equivalent diffusion key $K_{d}$, the structure of the two-round equivalent encryption algorithm shown in Figure 2 is degenerated to the one-round permutation encryption structure. It is noted that the two-round permutation encryption algorithm is only used to change the coordinate position of pixels without changing the pixel value at the coordinate position, which can be equivalent to one round of global permutation. Therefore, the equivalent permutation key $K_{s}$ can be solved with the help of the position difference of special plaintext–ciphertext pairs. According to the analysis, the chosen plaintext attack can be used to crack the equivalent permutation key $K_{s}$, namely by constructing $N_{c}=\left\lceil \log_{L}\left(H\times W\right)\right\rceil$ plaintext images and their corresponding ciphertext image, the symbol ⌈·⌉ indicates the rounding-up operation, *H* and *W* are the height and width of the plaintext image, respectively, *L* is the total number of pixel values of all possible plaintext images, and for the 8-bit plaintext image, L=256.

#### 3.3.1. The Construction Method of $N_{c}$ Plaintext Images

The steps of constructing $N_{c}$ plaintext images by the chosen plaintext attack are as follows:

**Step 1.** Construct a special plaintext image $Q={\left\{Q\left(i,j\right)\right\}}_{i=1,j=1}^{H,W}$ with the same size as the original plaintext image $P={\left\{P\left(i,j\right)\right\}}_{i=1,j=1}^{H,W}$, and write the non-negative integers 0,1,2,⋯, and H×W−1 sequentially by scanning in the raster order (from left to right and top to bottom), that is the pixel value of the special plaintext image *Q* is Q(i,j)=(i−1)×W+(j−1), where i=1,2,⋯,H and j=1,2,⋯,W. Since the range of the pixel values is [0,L−1], *Q* needs to be decomposed into $N_{c}$ sub-matrices with pixel values in the range of [0,L−1].
$$Q=\left[\begin{array}{cccc} 0 & 1 & \cdots & W-1\\ W & W+1 & \cdots & 2W-1\\ \vdots & \vdots & \cdots & \vdots \\ (H-1)\times W & (H-1)\times W+1 & \cdots & H\times W-1 \end{array}\right].$$

**Step 2.** Calculate the number of special plaintext image required $N_{c}=\left\lceil \log_{L}\left(H\times W\right)\right\rceil$, and build $N_{c}$ special plaintext images as $Q_{1}$, $Q_{2}$, ⋯, $Q_{N_{c}}$.

**Step 3.** The pixel values of $N_{c}$ plaintext images $Q_{1}$, $Q_{2}$, ⋯, $Q_{N_{c}}$ are written by using the constructed special plaintext image *Q*. According to the mode of raster scanning, the writing rule of the *l*-th plaintext image $Q_{l}$ is
(26)$$\begin{array}{c} Q_{l}=\mathrm{mod}\left(\left\lfloor Q/L^{l-1}\right\rfloor ,L\right), \end{array}$$
where $l=1,2,\cdots ,N_{c}$, L=256, and ⌊·⌋ represents the rounding-down operation.

#### 3.3.2. The Equivalent Permutation Key

Based on the special plaintext image constructed in Section 3.3.1, the steps to crack the equivalent permutation key are as follows:

**Step 1.** Create $N_{c}$ special plaintext images $Q_{l}={\left\{Q_{l}\left(i,j\right)\right\}}_{i=1,j=1}^{H,W}$ ($l=1,2,\cdots ,N_{c}$) according to the method in Section 3.3.1. According to the chosen plaintext attack, these $N_{c}$ special plaintext images are encrypted, respectively, and the corresponding ciphertext is $C_{l}={\left\{C_{l}\left(i,j\right)\right\}}_{i=1,j=1}^{H,W}$ ($l=1,2,\cdots ,N_{c}$).

**Step 2.** $N_{c}$ ciphertext images $C_{l}={\left\{C_{l}\left(i,j\right)\right\}}_{i=1,j=1}^{H,W}$ ($l=1,2,\cdots ,N_{c}$) are processed by the equivalent diffusion key $K_{d}$ for backward-diffusion to offset the effect of diffusion encryption, and $N_{c}$ intermediate ciphertext images $I_{l}={\left\{I_{l}\left(i,j\right)\right\}}_{i=1,j=1}^{H,W}$ ($l=1,2,\cdots ,N_{c}$) are obtained.

**Step 3.** Combine $N_{c}$ intermediate ciphertext images $I_{l}={\left\{I_{l}\left(i,j\right)\right\}}_{i=1,j=1}^{H,W}$ ($l=1,2,\cdots ,N_{c}$) into a permutation matrix $Q_{index}$, and obtain
(27)$$\begin{array}{c} Q_{index}=\sum_{l=1}^{N_{c}}\left(I_{l}\times L^{l-1}\right), \end{array}$$
where $l=1,2,\cdots ,N_{c}$ and L=256.

**Step 4.** By comparing the position difference with the same pixel value in the special plaintext matrix *Q* and the permutation-only matrix $Q_{index}$, the equivalent permutation key of the global permutation encryption link can be obtained.

According to the above cracking steps, the detailed cracking process of equivalent permutation key $K_{s}$ is shown in Algorithm 3.
**Algorithm 3** The procedure of cracking the equivalent permutation key $K_{s}$.**Input:** 
*Q***Output:** 
$K_{s}$1:[H,W]=size(C)2:$N_{c}=\mathrm{ceil}\left(\log_{256}\left(H\times W\right)\right)$3:$K_{s}\leftarrow \mathrm{zeros}\left(H,W\right)$4:$Q_{l}\leftarrow \mathrm{mod}\left(\mathrm{floor}\left(Q/L^{l-1}\right),256\right)$5:**for** l=1 to $N_{c}$ **do**6:   $Q_{l}\leftarrow \mathrm{Encrypt}\left(Q_{l}\right)$7:**end for**8:**for** l=1 to $N_{c}$ **do**9:   $F_{l}\left(H,W\right)\leftarrow \mathrm{mod}\left(C_{l}\left(H,W\right)-K_{d}\left(H,W\right),256\right)$10:   **for** i=W−1 to 1 **do**11:     $F_{l}\left(H,W\right)\leftarrow \mathrm{mod}\left(C_{l}\left(H,j\right)-K_{d}\left(H,j\right)-C_{l}\left(H,j+1\right),256\right)$12:   **end for**13:   **for** i=H−1 to 1 **do**14:     $F_{l}\left(i,W\right)\leftarrow \mathrm{mod}\left(C_{l}\left(i,W\right)-K_{d}\left(i,W\right)-C_{l}\left(i+1,W\right),256\right)$15:   **end for**16:   **for** i=H−1 to 1 **do**17:     **for** j=W−1 to 1 **do**18:        $F_{l}\left(i,j\right)\leftarrow \mathrm{mod}\left(C_{l}\left(i,j\right)-K_{d}\left(i,j\right)-C_{l}\left(i,j+1\right)-C_{l}\left(i+1,j\right),256\right)$19:     **end for**20:   **end for**21:   $I_{l}\leftarrow \mathrm{mod}\left(F_{l}\left(1,1\right),256\right)$22:   **for** j=2 to *W* **do**23:     $I_{l}\left(1,j\right)\leftarrow \mathrm{mod}\left(F_{l}\left(1,j\right)-F_{l}\left(1,j-1\right),256\right)$24:   **end for**25:   **for** i=2 to *H* **do**26:     $I_{l}\left(i,1\right)\leftarrow \mathrm{mod}\left(F_{l}\left(i,1\right)-F_{l}\left(i-1,1\right),256\right)$27:   **end for**28:   **for** i=2 to *H* **do**29:     **for** j=2 to *W* **do**30:        $I_{l}\left(i,j\right)\leftarrow \mathrm{mod}\left(F_{l}\left(i,j\right)-F_{l}\left(i-1,j\right)-F_{l}\left(i,j-1\right),256\right)$31:     **end for**32:   **end for**33:**end for**34:$Q_{index}\leftarrow \sum_{l=1}^{N_{c}}\left(I_{l}\times 256^{l-1}\right)$35:$K_{s}\leftarrow \mathrm{compare}\left(Q,Q_{index}\right)$36:**return** $K_{s}$

#### 3.3.3. Cracking Permutation Encryption Process by Using Equivalent Permutation Key

Based on the equivalent permutation key $K_{s}$ obtained in Section 3.3.2, the two-round permutation process is equivalent to one round of global permutation, and the attacker can recover the corresponding plaintext image from any given permutation image on the premise of unknown key parameters. The specific process is shown in Algorithm 4.

According to the whole process of cryptographic security analysis, for a plaintext image with a size of H×W, using the chosen plaintext attack only requires $1+\lceil \log_{L}\left(H\times W\right)\rceil$ special plaintext images, and the corresponding ciphertext images can crack the encryption algorithm.
**Algorithm 4** The procedure of cracking the permutation process by using equivalent permutation key $K_{s}$.**Input:** 
*I*, $K_{s}$**Output:** 
*P*1:[H,W]=size(I)2:seq_P←zeros(1,H×W)3:$seq\_I\leftarrow \mathrm{reshape}(I^{{}^{\prime }},1,H\times W)$4:$seq\_K_{s}\leftarrow \mathrm{reshape}\left(K_{s}^{{}^{\prime }},1,H\times W\right)$5:**for** i=1 to H×W **do**6:   $seq\_P(seq\_K_{s}\left(i\right))=seq\_I\left(i\right)$7:**end for**8:$P\leftarrow \mathrm{reshape}{\left(seq\_P,W,H\right)}^{{}^{\prime }}$9:**return** *P*

## 4. Experimental Results

The experimental hardware platform was a PC equipped with an Intel(R) Core(TM) i7 processor; the main frequency was 2.59 GHz; the memory (RAM) was 16 GB; the operating system was the Windows10 Professional 64-bit operating system; the software was MATLAB R2020a. In order to be consistent with the experiment of the original encryption algorithm on the dataset, the key parameters were selected from the original encryption algorithm, namely h=5, r=5, x(0)=0.5, and y(0)=0.5. Meanwhile, the same images from the USC-SIPI image database [39] were selected in the experiment. Here, two plaintext gray images, 5.1.09 and 5.1.10, with a size of 256×256, were selected as the test images. In order to facilitate a unified description, they were named according to the *Description* in the image database, namely Moon surface and Aerial. In addition, in order to verify the reliability of the cracking algorithm in this paper, test diagrams of different sizes should be selected. In this paper, Lena and cameraman with a size of 512×512 were selected for the experimental testing.

### 4.1. Experimental Results of Cracking Image by Chosen Plaintext Attack

According to the chosen plaintext attack method proposed in this paper, numerical experiments were carried out on the plaintext images with different sizes of 256×256 and 512×512. First of all, for the diffusion process, no matter how big the image size is, according to the analysis, it only needs to select an all-zero plaintext image $P_{0}={\left\{P_{0}\left(i,j\right)\right\}}_{i=1,j=1}^{H,W}=0$ and obtain its corresponding ciphertext image $C_{0}={\left\{C_{0}\left(i,j\right)\right\}}_{i=1,j=1}^{H,W}$ through encryption, as shown in Figure 5.

Secondly, for the permutation process, according to the special plaintext construction method proposed in Section 3.3.1, for images with a size of 256×256, the number of plaintexts to be constructed is $N_{c}=\left\lceil \log_{256}\left(256\times 256\right)\right\rceil =2$. Therefore, to crack the permutation process of images with a size of 256×256, it is necessary to select two special plaintext images $Q_{1}$ and $Q_{2}$ and obtain their corresponding ciphertext images $C_{1}$ and $C_{2}$ through encryption, as shown in Figure 6.

For a 512×512 image, the number of plaintexts that need to be constructed is $N_{c}=\left\lceil \log_{256}\left(512\times 512\right)\right\rceil =3$. Therefore, to crack the permutation process of 512×512 images, three special plaintexts $Q_{3}$, $Q_{4}$, and $Q_{5}$ need to be selected, whose corresponding ciphertext is $C_{3}$, $C_{4}$, and $C_{5}$, as shown in Figure 7.

Finally, the equivalent permutation key and equivalent diffusion key were used to crack and recover the plaintext image from the ciphertext image. For the plaintext gray images Moon surface and Aerial with a size of 256×256, the test results are shown in Figure 8. This includes the original gray image, the ciphertext image, the permutation-only image after cracking the diffusion process, and the equivalent cracking plaintext image.

For the plaintext gray images Lena and cameraman with a size of 512×512, the corresponding plaintext image can also be completely recovered from the ciphertext image. The results of the cracking test are shown in Figure 9, including the original gray image, the ciphertext image, the permutation-only image after the cracking diffusion process, and the equivalent cracked plaintext image.

### 4.2. Suggestions for Improvement

According to the security analysis of this paper, the original encryption algorithm had security defects and could not withstand the chosen plaintext attack. The following suggestions are given to fix the loopholes of the original encryption algorithm in the process of cryptanalysis:

(1) The two-round diffusion process should avoid operation composition. The iterative analysis process of forward-diffusion and backward-diffusion is actually the process of operation compounding, which makes the two rounds of diffusion able to be equivalent to one round of global diffusion, and its equivalent diffusion key can be cracked by the analysis of only one plaintext image. In the design process of the encryption algorithm, the operation compounding should be avoided, so as to avoid the invalidation of the round number for diffusion.

(2) The permutation and diffusion structure of the original algorithm should be improved. Although the permutation and diffusion processes of the original encryption algorithm are relatively complex, the structure of the whole encryption algorithm is relatively simple, which makes the whole structure equivalent to a single-round permutation–diffusion structure. It is suggested to improve the complexity of the algorithm structure to make all parts closely connected, thereby improving the security of the encryption algorithm.

## 5. Comparison and Performance Analysis of Cracking Algorithms

According to the previous analysis, the attack method proposed in this paper can efficiently and quickly obtain the equivalent key and then effectively crack the original encryption algorithm. This paper made a comprehensive comparison of the chosen plaintext attack methods proposed in [40] and analyzed the differences in their performance.

### 5.1. Comparison of Cracking Diffusion Processes

In [40], for the diffusion part, according to the similarity of the two rounds of the diffusion formula structure, a conclusion is given: the pixel value after diffusion was obtained modulo 256 by the sum of the pixel value before diffusion and the element value of the equivalent key matrix, that is
(28)$$\begin{array}{c} C\left(i,j\right)=\mathrm{mod}(f_{p}\left(i,j\right)+K\left(i,j\right),256), \end{array}$$
where $f_{p}\left(i,j\right)$ represents the variable related to the plaintext and its position and K(i,j) represents the equivalent diffusion key related to the diffusion key.

According to Equation (Equation 28), all the values related to the diffusion key in the two-round diffusion formula can be extracted and made uniformly equivalent with an equivalent diffusion key. In this case, there is no quantity related to the diffusion key in the two-round diffusion formula, so the two-round diffusion process will be weakened into an ordinary equation operation. The diffusion formula after weakening is as follows.

For forward-diffusion,
(29)$$S\left(i,j\right)=\left\{\begin{array}{cc} \mathrm{mod}(I(1,1),256) & \mathrm{if}\;i=1,j=1,\\ \mathrm{mod}(I(1,j)+S(1,j-1),256) & \mathrm{if}\;i=1,1<j\leq W,\\ \mathrm{mod}(I(i,1)+S(i-1,1),256) & \mathrm{if}\;1<i\leq H,j=1,\\ \mathrm{mod}(I(i,j)+S(i,j-1)+S(i-1,j),256) & \mathrm{if}\;1<i\leq H,1<j\leq W, \end{array}\right.$$
where i=1,2,⋯,H and j=1,2,⋯,W.

For backward-diffusion,
(30)$$C^{{}^{\prime }}\left(i,j\right)=\left\{\begin{array}{cc} \mathrm{mod}(S(H,W),256) & \mathrm{if}\;i=H,j=W,\\ \mathrm{mod}(S\left(H,j\right)+C^{{}^{\prime }}\left(H,j+1\right),256) & \mathrm{if}\;i=H,1\leq j<W,\\ \mathrm{mod}(S\left(i,W\right)+C^{{}^{\prime }}\left(i+1,W\right),256) & \mathrm{if}\;1\leq i<H,j=W,\\ \mathrm{mod}(S\left(i,j\right)+C^{{}^{\prime }}\left(i,j+1\right)+C^{{}^{\prime }}\left(i+1,j\right),256) & \mathrm{if}\;1\leq i<H,1\leq j<W, \end{array}\right.$$
where i=H,H−1,⋯,1 and j=W,W−1,⋯,1.

According to the above analysis, the whole diffusion process can be equivalent to the structure shown in Figure 10.

The equivalent processing method is obviously different from the equivalent processing method used in this paper. The equivalent structure in this paper is shown in Figure 2. The above equivalent processing method skillfully utilizes the similarity of the structure of the two rounds of diffusion formulas and combines the properties of the mathematical operations to extract the two diffusion keys into an equivalent diffusion key *K*, so as to crack the original encryption algorithm. However, if the structure of the two-round diffusion formula is not similar, the method will fail, so it is not universal. The method proposed in this paper does not need to extract the two equivalent keys, but combines the two rounds of the diffusion process by eliminating the intermediate ciphertext *S*, which is equivalent to the classical round of the permutation–diffusion structure, and the analysis of this kind of encryption algorithm can be widely applied.

### 5.2. Comparison of Cracking Permutation Process

In [40], the method of the chosen plaintext attack was adopted for the cracking of the permutation process. Firstly, according to the analysis, it can be known that, for an image with a size of H×W and the same *L* different pixel values, $N_{c}\geq \log_{L}\left(H\times W\right)$ special plaintext–ciphertext pairs are required to obtain the equivalent diffusion key.

In order to illustrate the above process of selecting special information to crack the permutation key, a 2×2 matrix is used to demonstrate the process of obtaining the equivalent permutation key.

For the case of H=W=L=2, a matrix with element values increasing from 0 should be constructed according to the *L*-base (binary here), then decomposed into $N_{c}$ special plaintext bit by bit. The corresponding ciphertext images should be obtained by permutation encryption, and the pixel values in plaintext–ciphertext pairs before and after permutation encryption are compared. Analyze and judge the possible source of pixel value of each position. Finally, it is necessary to take the intersection of all possible situations in the $N_{c}$ matrix to obtain the equivalent permutation key.

The above method to crack the equivalent permutation key has a big defect. When comparing the pixel values in plaintext–ciphertext pairs before and after permutation encryption and analyzing and determining the possible source of the pixel value at each position, all pixels of the entire image matrix need to be traversed repeatedly, which requires a large amount of memory resources and consumes a long time, which greatly reduces the entire cracking efficiency. According to the simulation experiment results in [40], it takes nearly 2 h to crack images with a size of 256×256, and the time consumed increases exponentially with the increase of the image size.

In contrast, with the method proposed in this paper, it is unnecessary to repeatedly traverse all the pixel values in the image, so the equivalent permutation key can be obtained efficiently without increasing the number of special plaintexts, and it only takes 0.63178 s to crack an image of size 256×256, while it takes 8.03338 s to crack an image of size 512×512. The performance of the proposed attack method has higher superiority.

### 5.3. Comparison of Attack Complexity

The attack complexity mainly includes the time complexity and the data complexity. The following is a comparison of cracking methods from these two aspects.

In terms of the data complexity, according to the attack method proposed in this paper, the number of plaintext–ciphertext pairs required to crack the entire encryption algorithm is $1+\lceil \log_{256}\left(H\times W\right)\rceil$, while the attack method in [40] requires at least the same number of plaintext–ciphertext pairs. Therefore, the data complexity of both decoding algorithms is O(log(H×W)).

In terms of the time complexity, according to the test for pictures of different sizes in [40], the same tests were also conducted for the cracking algorithm proposed in this paper. For the convenience of comparison, the test images in the USC-SIPI image dataset provided in [40] were uniformly used. In addition, in order to ensure the accuracy of the data, the experimental data in [40] are directly quoted here. At the same time, the experimental hardware platform with the same configuration (equipped with an Intel (R) Core (TM) i7 processor; the main frequency was 2.59 GHz; the memory was 16 GB) was used to crack images of different sizes many times, and multiple sets of test results were obtained. Finally, the average value of the test results was taken as the final test results. The running time is shown in Table 1.

The broken line graphs in Figure 11, respectively, show the growth trend of the running time of the two cracking methods. With the increase of the size of the test image, the running time of both cracking algorithms grew. However, the running time of this paper was generally far lower than that of the cracking algorithm in [40]. Therefore, the cracking algorithm proposed in this paper had a high running efficiency and could recover the plaintext image at a very fast speed without increasing the data complexity.

## 6. Conclusions

In this paper, the security of an image encryption algorithm based on a two-dimensional hyperchaotic map was analyzed in detail. Through the security analysis, it was found that the encryption algorithm cannot withstand the chosen plaintext attack. The key used in the two-round permutation and two-round diffusion of the original encryption algorithm was independent of the plaintext image. Through further theoretical derivation and analysis, it was found that there was an equivalent key for this algorithm structure, which can be simplified into the one-round global permutation and one-round global diffusion structure. Therefore, the method of the chosen plaintext attack was proposed to crack the algorithm. Theoretical analysis and numerical simulation results showed that, according to the chosen plaintext attack method proposed in this paper, for the plaintext gray image with a size of H×W, only $1+\lceil \log_{256}\left(H\times W\right)\rceil$ special plaintext images and their corresponding ciphertext images needed to be selected to obtain the equivalent permutation and diffusion key, so as to realize the cracking of the original encryption algorithm. In the analysis method of this paper, the number of special plaintext images needed to crack the original encryption algorithm was small, and the attack complexity was not high, while the effectiveness of the attack algorithm was verified by the simulation test. Compared with the existing cracking algorithms, the chosen plaintext attack method in this paper consumed less time and was more efficient in cracking the original encryption algorithm without increasing the data complexity.

## Figures and Tables

**Figure 1 entropy-25-00395-f001:**
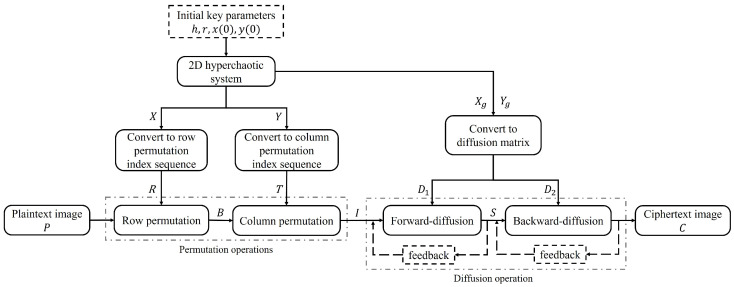
The block diagram of the original encryption algorithm.

**Figure 2 entropy-25-00395-f002:**
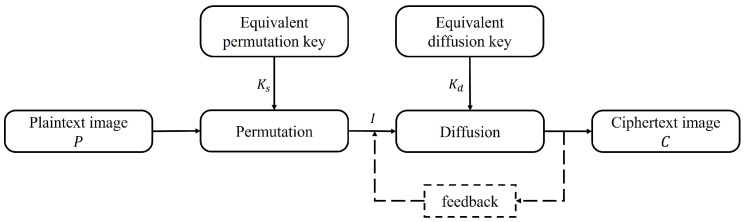
The block diagram of equivalent encryption algorithm.

**Figure 3 entropy-25-00395-f003:**
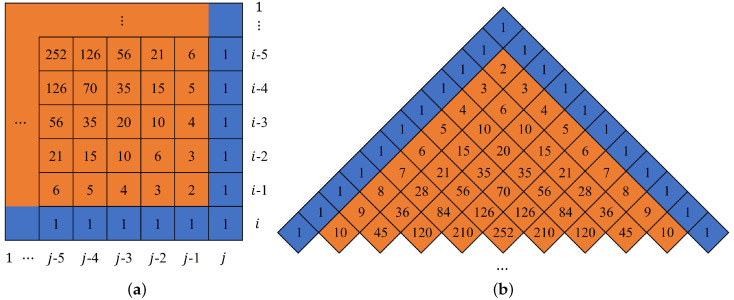
Matrix *A* and its corresponding value. (**a**) Matrix *A*; (**b**) Pascal’s triangle.

**Figure 4 entropy-25-00395-f004:**
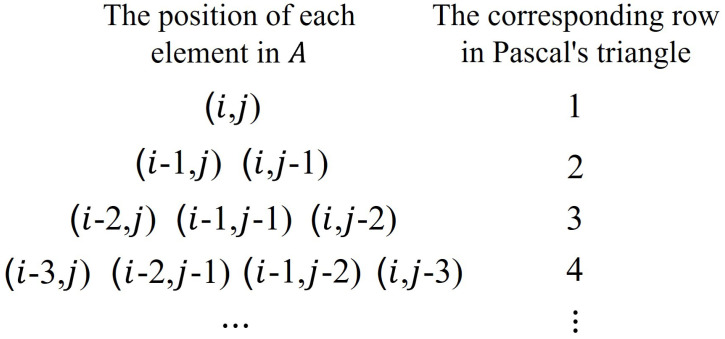
The corresponding relationship between the values in matrix *A* and Pascal’s triangle.

**Figure 5 entropy-25-00395-f005:**
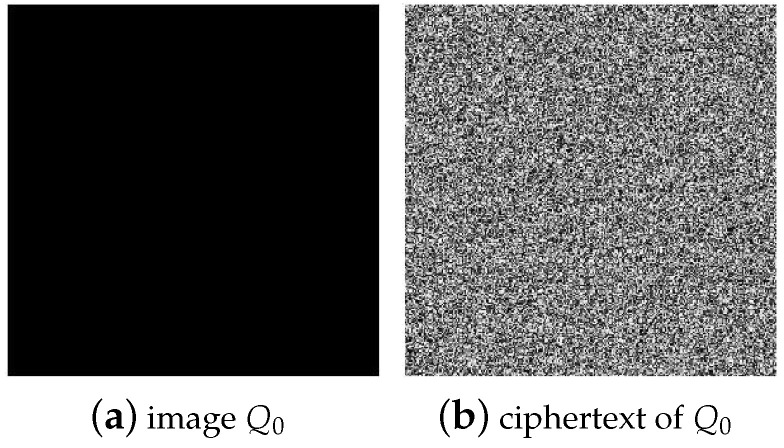
All-zero plaintext image and its corresponding ciphertext image: (**a**) the all-zero plaintext image $P_{0}$; (**b**) the corresponding ciphertext image $C_{0}$ of $P_{0}$.

**Figure 6 entropy-25-00395-f006:**
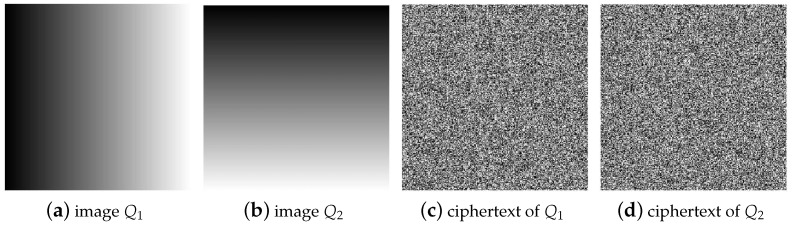
Select special plaintext images with a size of 256×256 and their corresponding ciphertext images: (**a**) the special plaintext image $Q_{1}$; (**b**) the special plaintext image $Q_{2}$; (**c**) the corresponding ciphertext image $C_{1}$ of $Q_{1}$; (**d**) the corresponding ciphertext image $C_{2}$ of $Q_{2}$.

**Figure 7 entropy-25-00395-f007:**
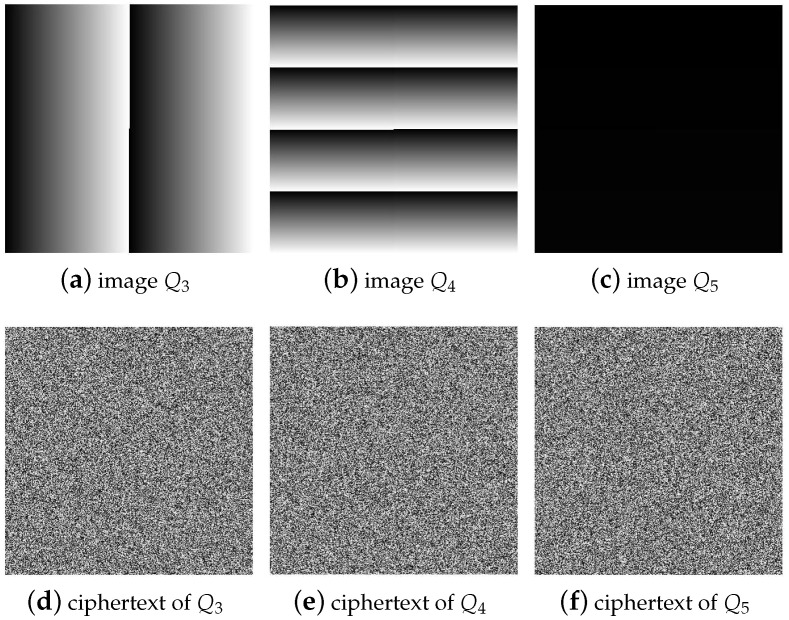
Select special plaintext images with a size of 512×512 and their corresponding ciphertext images: (**a**) the special plaintext image $Q_{3}$; (**b**) the special plaintext image $Q_{4}$; (**c**) the special plaintext image $Q_{5}$; (**d**) the corresponding ciphertext image $C_{3}$ of $Q_{3}$; (**e**) the corresponding ciphertext image $C_{4}$ of $Q_{4}$; (**f**) the corresponding ciphertext image $C_{5}$ of $Q_{5}$.

**Figure 8 entropy-25-00395-f008:**
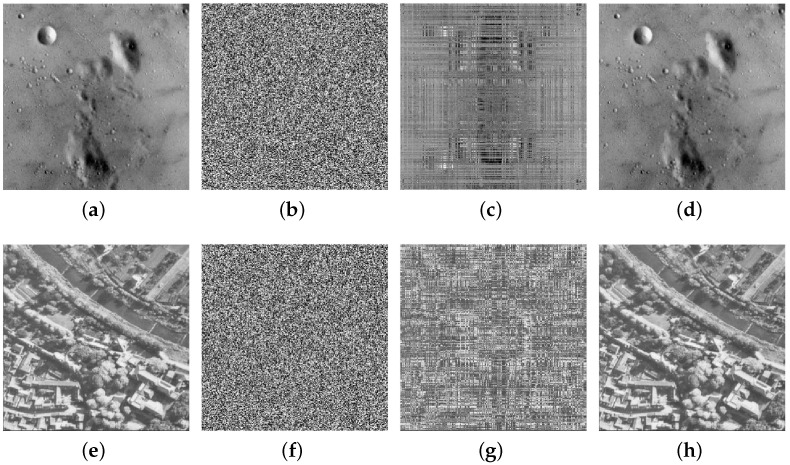
Cracking results for plaintext gray images of a size 256×256: (**a**) the original gray image of the Moon’s surface; (**b**) the ciphertext image of the Moon’s surface; (**c**) the cracked Moon’s surface permutation image; (**d**) the cracked plaintext image of the Moon’s surface; (**e**) the original gray image of Aerial; (**f**) the ciphertext image of Aerial; (**g**) the cracked Aerial permutation image; (**h**) the cracked plaintext image of Aerial.

**Figure 9 entropy-25-00395-f009:**
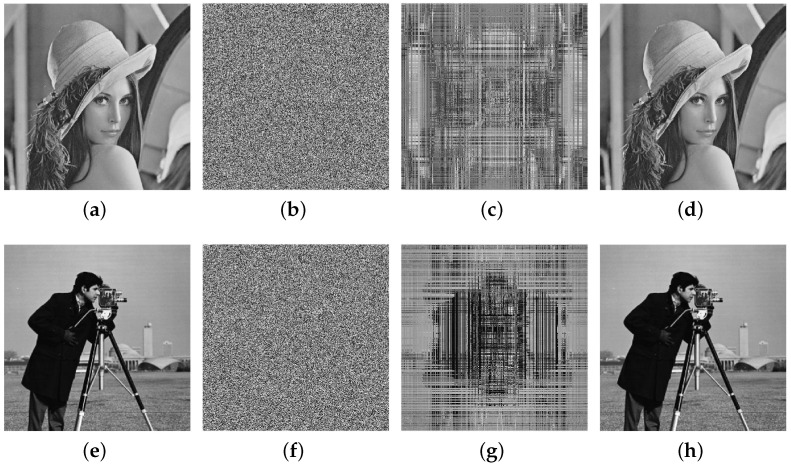
Cracking results for plaintext gray images of a size 512×512: (**a**) the original gray image of Lena; (**b**) the ciphertext image of Lena; (**c**) the cracked Lena permutation image; (**d**) the cracked plaintext image of Lena; (**e**) the original gray image of the cameraman; (**f**) the ciphertext image of the cameraman; (**g**) the cracked cameraman permutation image; (**h**) the cracked plaintext image of the cameraman.

**Figure 10 entropy-25-00395-f010:**
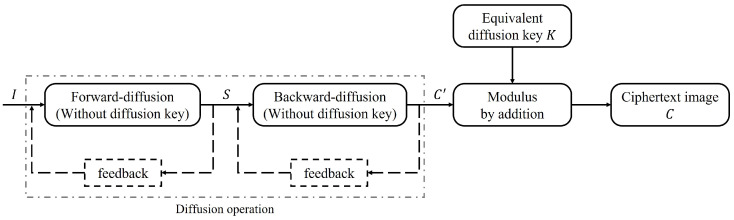
Diffusion structure after equivalence.

**Figure 11 entropy-25-00395-f011:**
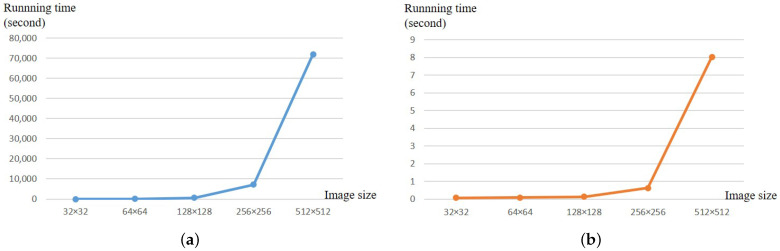
The running time of the two cracking algorithms: (**a**) the cracking algorithm in [40]; (**b**) our cracking algorithm.

**Table 1 entropy-25-00395-t001:** The running time of the two cracking algorithms.

Image Size	32 × 32	64 × 64	128 × 128	256 × 256	512 × 512
Running time in [40]/s	8	50	540	7200	72,000
Running time in this paper/s	0.08396	0.09188	0.13891	0.63178	8.03338

## Data Availability

Not applicable.

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
