# Peer review of "Cryptanalysis of an Image Encryption Algorithm Based on Two-Dimensional Hyperchaotic Map"

_entropy, 2023, doi:10.3390/e25030395_

Round 1

Reviewer 1 Report

In this paper, the security of the image encryption algorithm based on two-dimensional hyperchaotic map is analyzed, and the chosen plaintext attack is used to crack the diffusion and permutation respectively, so that the images of different sizes can be recovered with lower attack complexity. The logic of the algorithm analysis process is clear, and the corresponding pseudo code is given at each stage. The effectiveness of the cracking algorithm is tested through experiments. Finally, compared with the existing decryption algorithms, the running time of the cracking algorithm is greatly reduced, and the efficiency is significantly improved without increasing the data complexity, through theoretical analysis and experimental verification. In response to the existing questions, the following comments and suggestions are put forward:

 1. The paper discusses the improved encryption algorithm based on permutation and diffusion structure in recent years. In my opinion, the improved encryption algorithm based on chaotic map can also be mentioned in the introduction, as well as the practical application in related fields. Therefor the authors may consult the following references papers (not necessarily to be cited).

(i)    A Memristive Synapse Control Method to Generate Diversified Multi-Structure Chaotic Attractors.

(ii) Generating n-scroll chaotic attractors from a memristor-based magnetized hopfield neural network.

(iii)        Brain-like initial-boosted hyperchaos and application in biomedical image encryption.

(iv)  Hyperchaotic memristive ring neural network and application in medical image encryption.

(v)   A novel hyper-chaotic image encryption scheme based on quantum genetic algorithm and compressive sensing.

2. The proof object in Proposition 1 is mathematically similar to the Pascal matrix. Is there a connection between the two? Are there any conclusions that can be directly quoted?

 3. What are the advantages of using this method compared with existing cracking algorithms?

 4. For the comparison experiment part of attack complexity, whether a unified test image is used for images of different sizes, and how to ensure the accuracy of the test results?

5. Compared with the existing algorithm, can you explain the reason from the theoretical analysis for the gap between the operation efficiency shown in the experimental results?

Reviewer 2 Report

1. Lines 60-61 page 2:

   "Among many cryptanalysis methods, chosen plaintext attack is a commonly used

   method. For example, the equivalent key can be obtained by selecting the corresponding"

   Please, explain briefly the "chosen plaintex attack"

2. Lines 199-200 in page 7:

   ( p = i − k + j − l + 1) of Pascal’s triangle. The first element of the p-th row of Pascal’s

   triangle is a(i − p + 1, j), the row number of the element a increases with a tolerance of

   What is 'a' in the expression "a(i − p + 1, j)"?

3. Lines 330-331, page 17:

  the experiment are also the same as those in [34]. Here, two plaintext gray images 5.1.09

  and 5.1.10 with a size of 256 × 256 are selected as the test images, which correspond to the

  Why the gray images are named as "5.1.09" and "5.1.10" and not as simply image '1' and image '2'?

4. Please, add labels "image Q1", "image Q2", "ciphertext of Q1", and "ciphertext of Q2"

   to (a), (b), (c), and (d) subcaption in Figure 6.

5. Ok, something I don't understand clearly from the proposed  method in the document:

   a) The key in which is supported all the security is the value of the

       initial key.

      This initial key is the set of values for x(0), y(0), h, and r

      in Eq. (1). The security of the algorithm is how an adversary

      can guess theses used values from an encrypted image knowing

      in advance the encryption and decryption algorithm.

   b) In the proposed method  described in the paper, based in the chosen

      plain attack, the same set of values for x(0), y(0), h, and r

      is used under all the experiments. Then, it is equivalent to

      known in advance the secure key of the entire process.

   c) If the idea on my point (b) is not correct, authors must

      describe which is the correct idea. Or equivalent, how

      the set of values x(0), y(0), h, and r can be recovered (the key)

      from a given encrypted image?

Round 2

Reviewer 1 Report

The revised paper has addressed the questions proposed by reviewers, and can be published.

Reviewer 2 Report

Authors did my suggested changes to the document. Now the paper can be accepted.